# Synergistic Radiation Effects in PPD CMOS Image Sensors Induced by Neutron Displacement Damage and Gamma Ionization Damage

**DOI:** 10.3390/s24051441

**Published:** 2024-02-23

**Authors:** Zu-Jun Wang, Yuan-Yuan Xue, Ning Tang, Gang Huang, Xu Nie, Shan-Kun Lai, Bao-Ping He, Wu-Ying Ma, Jiang-Kun Sheng, Shi-Long Gou

**Affiliations:** 1National Laboratory of Intense Pulsed Irradiation Simulation and Effect, Northwest Institute of Nuclear Technology, Xi’an 710024, Chinahebaoping@nint.ac.cn (B.-P.H.); mawuying@nint.ac.cn (W.-Y.M.); shengjiangkun@nint.ac.cn (J.-K.S.); goushilong@nint.ac.cn (S.-L.G.); 2School of Materials Science and Engineering, Xiangtan University, Xiangtan 411105, China

**Keywords:** CMOS image sensor (CIS), dark signal, dark signal non-uniformity (DSNU), displacement damage, gamma ray, neutron, radiation, synergistic effect, total ionizing dose (TID)

## Abstract

The synergistic effects on the 0.18 µm PPD CISs induced by neutron displacement damage and gamma ionization damage are investigated. The typical characterizations of the CISs induced by the neutron displacement damage and gamma ionization damage are presented separately. The CISs are irradiated by reactor neutron beams up to 1 × 10^11^ n/cm^2^ (1 MeV neutron equivalent fluence) and ^60^Co γ-rays up to the total ionizing dose level of 200 krad(Si) with different sequential order. The experimental results show that the mean dark signal increase in the CISs induced by reactor neutron radiation has not been influenced by previous ^60^Co γ-ray radiation. However, the mean dark signal increase in the CISs induced by ^60^Co γ-ray radiation has been remarkably influenced by previous reactor neutron radiation. The synergistic effects on the PPD CISs are discussed by combining the experimental results and the TCAD simulation results of radiation damage.

## 1. Introduction

Complementary metal oxide semiconductor image sensors (CISs) have been widely used in the nuclear industry, space applications, medical imaging, and particle detection as the detection or imaging devices for their advantages of high sensitivity, high levels of integration, low cost, and low power operation [1,2,3,4,5]. However, the CISs used in the above applications will be operated in radiation environments and be susceptible to radiation damages such as total ionizing dose (TID) damage, displacement damage, and transient radiation damage [1,2,3,4,5,6,7,8,9,10,11,12,13,14].

Many studies have been published on the radiation effects on the pinned photodiode (PPD) CISs [6,7,8,9,10,11,12,13,14,15,16,17]. The literature is mainly focused on one of the radiation effects on the CISs such as total ionizing dose effects, displacement damage effects, and single event effects. The radiation-induced stable accumulated dose damage can be mainly separated into the TID damage and the displacement damage. However, fewer papers have focused on the synergistic radiation effects on the 0.18 µm PPD CISs induced by the TID damage and the displacement damage. Zujun Wang et al. studied radiation effects on 0.35 µm CMOS APS induced by combined reactor neutron beams and ^60^Co γ-rays in previous paper [18]. Considering the CISs of the current mainstream applications are mainly manufactured by a standard 0.18 µm CMOS process, many radiation experiments have been carried out to investigate the radiation effects on the 0.18 µm PPD CISs. The experimental results show that the degradations of the CISs induced by ionization damage are notable when they were exposed to the ^60^Co γ-rays at the TID level of 200 krad(Si) and the degradations of the CISs induced by displacement damage are also notable when they were exposed to the neutron beams up to 1 × 10^11^ n/cm^2^ (1 MeV neutron equivalent fluence). Based on the radiation experimental results, the experimental schemes have been improved to research the synergistic radiation effects on the 0.18 µm PPD CISs induced by displacement damage and ionization damage. We think that the synergistic radiation effects of the CISs induced by displacement damage and ionization damage will be easier to observe when both the displacement damage and the ionization damage are notable. The radiation experiments and annealing of neutron displacement damage and gamma ionization damage have been carried out because the neutron radiation mainly induces the displacement damage while the gamma radiation mainly induces the ionization damage.

The research reported herein examines the synergistic effects on the 0.18 µm PPD CISs induced by neutron displacement damage and gamma ionization damage. The typical characterization induced by gamma ionization damage effects and the annealing tests are presented in Section 3.1, and the typical characterization induced by neutron displacement damage and annealing are presented in Section 3.2. The experimental results of the CISs that have been firstly exposed to ^60^Co γ-rays and have been secondly exposed to neutron beams are presented in Section 3.3, and the experimental results of the CISs that have been firstly exposed to neutron beams and have been secondly exposed to ^60^Co γ-rays are presented in Section 3.4. Finally, we discuss the synergistic effects on the PPD CISs by combining the experimental results and the TCAD simulation results of radiation damage in Section 4. The research has verified the synergistic effects on the 0.18 µm PPD CISs induced by neutron displacement damage and gamma ionization damage.

## 2. Materials and Methods

### 2.1. Irradiation and Annealing

The devices under test (DUTs) were exposed to neutrons from a reactor and to gamma-rays from a ^60^Co source. The neutron radiation experiments of the CISs were carried out at Xi’an pulse reactor (XAPR) facility of Northwest Institute of Nuclear Technology (Xi’an, China). The neutron beam homogeneity was within ±10%. The flux of neutron beams was about 3.39 × 10^8^ n/(cm^2^ s), and was measured by staff at the radiation facility. The ratio of n/γ is 1.29 × 10^9^ cm^−2^/rad(Si). The gamma irradiation experiments of the CISs were carried out at the ^60^Co γ-ray resource of Northwest Institute of Nuclear Technology (Xi’an, China). The dosimetry is accurate to better than 2.5%. The dose rate of ^60^Co γ-rays is 50.0 rad(Si)/s.

Two samples (A01 and A02) were firstly exposed to ^60^Co γ-rays up to the TID level of 200 krad(Si) and then underwent a 168 h annealing at room temperature. The samples were secondly exposed to a neutron beam up to 1 × 10^11^ n/cm^2^ (1 MeV neutron equivalent fluence) and then underwent a 168 h annealing at room temperature. The neutron fluence referred to this paper is the 1 MeV neutron equivalent fluence. We cannot find the radiation resources that can provide the radiation environments of the TID level of 200 krad(Si) and neutron fluence level of 1 × 10^11^ n/cm^2^ at the same time. Therefore, the CISs are irradiated by ^60^Co γ-rays and reactor neutron beams with different sequential order. The radiation test conditions and samples are presented in Table 1.

The samples have the same lot date code. The samples are unbiased with all pins floating during all irradiations. All the samples were measured by a CMOS image sensor radiation test system. The parameters of the CMOS APS image sensors were measured before and after radiation.

Two samples (B01 and B02) were firstly exposed to a neutron beam up to 1 × 10^11^ n/cm^2^ and then underwent 168 h annealing at room temperature. The samples were secondly exposed to ^60^Co γ-rays up to the TID level of 200 krad(Si) and then underwent 168 h annealing at room temperature. The radiation test conditions and samples are presented in Table 2.

### 2.2. Tested CISs

The samples under tests are manufactured using a standard 0.18-μm CMOS technology with six-transistor (6T) pinned photodiode pixels. The PPD pixel includes the pre-metal dielectric (PMD), shallow trench isolation (STI), gate oxide, transfer gate (TG), and space charge region (SCR). The samples of the backside-illuminated scientific CISs have four millionpixels with an electronic rolling shutter. The frame readout image has 2048 × 2048 elements. The full well is 54 ke^−^. The pixel size is 6.5 × 6.5 μm^2^ and the dynamic range is 90 dB. The readout noise is 1.6 e^−^.

## 3. Results

### 3.1. Gamma Irradiation and Annealing

The CISs are very sensitive to the ionization damage. The typical phenomenon of the CISs induced by the ionization damage is the dark signal increase [1,2,3,4,5,6,7,8,14]. For example, the TID damage will induce the dark signal increase and the dark signal non-uniformity increase. In this paper, we are focused on the dark signal increase induced by radiation to research the synergistic effects on the PPD CISs.

Figure 1 shows the dark signal distributions induced by^60^Co γ-ray radiation and annealing at 24 h, 96 h, and 168 h with the same integration time of 16.79 ms. From Figure 1, one can see that the dark signal increases notably from the dark signal distributions induced by ^60^Co γ-ray radiation at 100 krad(Si) and 200 krad(Si). The dark signal degradations are mainly due to the trapped positive charges and the interface states induced by TID damage at the PMD, STI, TG, nitride spacer, and gate oxide.

The annealing tests at 24 h, 96 h, and 168 h have been carried out after the ^60^Co γ-ray radiation at 200 krad(Si). From the dark signal distributions after annealing shown in Figure 1, one can see that the dark signal decreases notably. Figure 2 shows the mean dark signal induced by ^60^Co γ-ray radiation and annealing at 24 h, 96 h, and 168 h versus integration time. From Figure 2, one can see that the mean dark signals increase with increasing integration time. The mean dark signals in the pixels of the CISs will accumulate with integration time. Before the dark signals are full of the pixel well to saturation, the longer the integration time, the larger the dark signal. After annealing for 168 h, the mean dark signals tend to be stable, which is due to the relative stability of the number of trapped positive charges and the interface states induced by the TID damage at the PMD, STI, TG, nitride spacer, and gate oxide.

To further understand the degradation characteristics of dark signal, the typical dark image induced by ^60^Co γ-ray radiation is presented. Figure 3 shows the typical dark images induced by ionization damage (inversion of black and white color): (a) before radiation; (b) 200 krad(Si) after ^60^Co γ-ray radiation. From Figure 3, one can see that the dark image has been degraded notably. The dark signal degradation of the dark image shows a uniform increase. This phenomenon is very different with that induced by neutron radiation.

### 3.2. Neutron Irradiation and Annealing

The CISs are also sensitive to the displacement damage. The typical phenomenon of the CISs induced by the displacement damage is the dark signal increase, the dark signal non-uniformity increase, and the dark signal spike increase. Figure 4 shows the dark signal distributions induced by reactor neutron radiation and annealing for 168 h with the same integration time of 16.79 ms. From Figure 4, one can see that the dark signal increases notably from the dark signal distributions induced by reactor neutron radiation at 1 × 10^11^ n/cm^2^. The equivalent TID is about 0.078 krad(Si) when the neutron fluence is 1 × 10^11^ n/cm^2^ as the ratio of n/γ referred in the Section 2. Thus, the radiation damage of the CISs induced by reactor neutron radiation at 1 × 10^11^ n/cm^2^ is mainly due to the displacement damage, and the ionization damage is negligible. The dark signal degradations are mainly due to bulk traps induced by the neutron displacement damage at the SCR, STI, and TG. The 168 h annealing tests have been carried out after the reactor neutron radiation at 1 × 10^11^ n/cm^2^. The dark signal shows no notable decrease from the dark signal distributions after 168 h annealing.

Figure 5 shows the mean dark signal induced by reactor neutron radiation and annealing for 168 h versus integration time. From Figure 5, one can see that the mean dark signals increase with increasing integration time. After annealing for 168 h, the mean dark signal shows a slight decrease, which is due to the relative stability of the number of the bulk traps induced by reactor neutron damage at the SCR, STI, and TG. The previous experimental results show that the annealing phenomenon induced by neutron radiation is not notable, which is very different to that induced by ^60^Co γ-ray radiation. After annealing for 168 h, we can see from Figure 5 that the mean dark signal decreases slightly at higher integration time, while it decreases notably at lower integration time, as shown in Figure 2.

Figure 6 shows the typical dark images induced by displacement damage (inversion of black and white color): (a) before radiation; (b) 1 × 10^11^ n/cm^2^ after reactor neutron radiation. From Figure 6, one can see that the dark image has been degraded notably. The dark signal degradation in the dark image induced by neutron radiation shows a local increase while the dark signal degradation in the dark image induced by ^60^Co γ-ray radiation shows a uniform increase.

### 3.3. Gamma Irradiation and Neutron Irradiation

From Section 3.1 and Section 3.2, we know about the typical characterization of the CISs induced by the neutron displacement damage and gamma ionization damage. In order to research the synergistic radiation effects in the CISs induced by displacement damage and ionization damage, the CISs were firstly exposed to ^60^Co γ-rays and then were secondly exposed to reactor neutron beams. Considering the fact that the dark signal degradations of the CISs induced by ^60^Co γ-ray radiation have a certain degree of recovery after annealing, we carried out a 168 h annealing at room temperature after ^60^Co γ-ray radiation at 200 krad(Si).

Figure 7 shows the dark signal distributions before and after radiation: before radiation (black line); ^60^Co γ-ray radiation at 200 krad(Si) after annealing for 168 h (red line); firstly exposed to ^60^Co γ-rays (including annealing for 168 h) and then secondly exposed to reactor neutron beams (blue line). From Figure 7, one can see that the dark signal distribution of the CISs induced by ^60^Co γ-rays and reactor neutrons shows the typical characterization of both the ionization damage and the displacement damage. The right part of the blue line in Figure 7 mainly shows the dark signal distribution of the CISs induced by the displacement damage while the left part mainly shows the dark signal distribution of the CISs induced by the ionization damage.

Figure 8 shows the mean dark signal degradations of the CISs that were firstly exposed to ^60^Co γ-rays and annealing for 168 h, and then were secondly exposed to reactor neutron beams and annealing for 168 h. All the tests are with the same integration time of 16.79 ms. From Figure 8, one can see that the mean dark signals of the CISs were degraded after radiation and annealing. The mean dark signal was 165.62 DN before radiation and was 2515.49 DN after ^60^Co γ-ray radiation at 200 krad(Si). Considering the annealing of the CISs after ^60^Co γ-ray radiation, a 168 h annealing test at room temperature has been carried out to keep the mean dark signal stable. The mean dark signal decreases from 2515.49 DN to 2112.53 DN after 168 h of annealing. The mean dark signal increases from 2112.53 DN to 2651.46 DN after reactor neutron radiation at 1 × 10^11^ n/cm^2^. The mean dark signal decreases from 2651.46 DN to 2338.66 DN after 168 h of annealing.

The experimental results show that the dark signal degradations of the CISs induced by ^60^Co γ-ray radiation or reactor neutron radiation have a certain degree of recovery after annealing though the dark signal degradations cannot be recovered to the level of the previous radiation. The mean dark signal increase is 2349.87 DN after ^60^Co γ-ray radiation at 200 krad(Si), while it is 538.93 DN after reactor neutron radiation at 1 × 10^11^ n/cm^2^. The experimental results in Figure 8 present the dark signal degradations induced by both the neutron displacement damage and ^60^Co γ-ray ionization damage.

### 3.4. Neutron Irradiation and Gamma Irradiation

From Section 3.3, we know about the typical characterization of the CISs that were firstly exposed to ^60^Co γ-rays and then were secondly exposed to reactor neutron beams. In this section, the experimental results of the CISs that were firstly exposed to neutron beams and were secondly exposed to ^60^Co γ-rays are presented.

Figure 9 shows the dark signal distributions before and after radiation: before radiation (black line); reactor neutron beams at 1 × 10^11^ n/cm^2^ after annealing for 168 h (red line); firstly exposed to reactor neutron beams(including annealing for 168 h) and then secondly exposed to ^60^Co γ-rays at 200 krad(Si) (blue line). From Figure 9, one can see that the dark signal distributions of the CISs induced by reactor neutrons and ^60^Co γ-rays also show the typical characterization of both the ionization damage and the displacement damage. However, the dark signal distributions in Figure 7 and Figure 9 are very different. This means that the degradations of the CISs are very different when the CISs are irradiated by reactor neutron beams and ^60^Co γ-rays with different sequential order.

For further quantitative analysis of experimental results, the mean dark signal degradation induced by neutron displacement damage and ^60^Co γ-ray ionization damage is presented. Figure 10 shows the mean dark signal degradations of the CISs which were firstly exposed to reactor neutron beams and annealing 168 h, and then were secondly exposed to ^60^Co γ-rays and annealing 168 h. All the tests are also with the same integration time of 16.79 ms. From Figure 10, one can see that the mean dark signals of the CISs were degraded after radiation and annealing.

The mean dark signal was 162.34 DN before radiation and was 693.99 DN after reactor neutron radiation at 1 × 10^11^ n/cm^2^. The mean dark signal decreases from 693.99 DN to 665.40 after 168 h of annealing. The mean dark signal increases from 665.40 DN to 1267.47 DN after ^60^Co γ-ray radiation at 200 krad(Si). The mean dark signal decreases from 1267.47 DN to 899.70 DN after 168 h of annealing. The experimental results show that the mean dark signal increase is 602.07 DN after ^60^Co γ-ray radiation at 200 krad(Si) while it is 531.65 DN after reactor neutron radiation at 1 × 10^11^ n/cm^2^.

The mean dark signal increase induced by reactor neutron radiation at 1 × 10^11^ n/cm^2^ as shown in Figure 10 is 531.65 DN (from 162.34 DN to 693.99 DN). The mean dark signal increase only induced by reactor neutron radiation at 1 × 10^11^ n/cm^2,^ as shown in Figure 8, is 538.93 DN (from 2112.53 DN to 2651.46 DN).This means that the mean dark signal increase in the CISs only induced by reactor neutron radiation at 1 × 10^11^ n/cm^2^ are very close, although there is one case that the CISs were firstly irradiated by ^60^Co γ-rays (including annealing 168 h) and then secondly irradiated by reactor neutron beams at 1 × 10^11^ n/cm^2^. The mean dark signal increase in the CISs induced by reactor neutron radiation has not been influenced by previous ^60^Co γ-ray radiation. However, the mean dark signal increasesin the CISs induced by ^60^Co γ-ray radiation at 200 krad(Si) as shown in Figure 8 and Figure 10 are very different. The mean dark signal increase induced by ^60^Co γ-ray radiation at 200 krad(Si) as shown in Figure 10 is 602.07 DN (from 665.40 DN to 1267.47 DN). The mean dark signal increase only induced by ^60^Co γ-ray radiation at 200 krad(Si)as shown in Figure 8 is 2349.87 DN (from 165.62 DN to 2515.49 DN). This means that the mean dark signal increase in the CISs induced by ^60^Co γ-ray radiation has been remarkably influenced by previous reactor neutron radiation. Therefore, we estimate there are synergistic effects on the 0.18 µm PPD CISs induced by neutron displacement damage and gamma ionization damage.

## 4. Discussion

From Section 3, we know about the typical characterization of the CISs induced by neutron displacement damage and gamma ionization damage. Based on the experimental results shown in Section 3, we have found that the dark signal degradations are very different when the CISs are irradiated by reactor neutron beams and ^60^Co γ-rays with different sequential order. Thus, we can discuss the synergistic effects on the 0.18 µm PPD CISs induced by neutron displacement damage and gamma ionization damage.

Figure 11 shows the comparison of the dark signal distributions of the CISs that were irradiated by reactor neutron beams and ^60^Co γ-rays with different sequential order. From Figure 11, one can see that the dark signal distribution of the CISs induced by reactor neutron beams and ^60^Co γ-rays with different sequential order is very different. The dark signal degradations of the CISs that were firstly exposed to ^60^Co γ-rays and annealing for 168 h, and then were secondly exposed to reactor neutron beams and annealing for 168 h are more serious. This phenomenon is consistent with the experimental results shown in Figure 8 and Figure 10.

Figure 12 shows the comparison of the dark signal increase in the CISs that were exposed to ^60^Co γ-rays (one case is that the CISs were firstly irradiated by reactor neutron beams at 1 × 10^11^ n/cm^2^ and annealing for 168 h, and then secondly irradiated by ^60^Co γ-rays). From Figure 12, one can see that the dark signal increase in the CISs induced by ^60^Co γ-ray radiation has been influenced by reactor neutron beams as description in Section 3.4. The dark signal degradations induced by ^60^Co γ-ray radiation decrease when the CISs were firstly irradiated by neutrons. In order to research the mechanisms of the above phenomenon, the TCAD simulation of the pixels in a CIS induced by neutron radiation was carried out. Figure 13 shows the electric field distribution of the region near the TG and SCR in a CIS: (a) before radiation; (b) after neutron radiation. From Figure 13, one can see that the electric field distribution of the region near the TG and SCR has changed after neutron radiation. The change in the electric field distribution will influence the recombination rates of the electron–hole pairs, which will influence the dark signal of the CISs.

Figure 14 shows the electric field of the region near the TG and SCR in a CIS before and after neutron radiation. From Figure 14, one can see that the electric field of the region near the TG and SCR decreases notably after neutron radiation. The dark signals induced by ^60^Co γ-ray radiation are correlated with the electric field. The electric field decrease induced by neutron radiation will increase the recombination rates of the electron–hole pairs, which induces the dark signal decrease.

## 5. Conclusions

In this paper, the synergistic effects on the 0.18 µm PPD CISs induced by neutron displacement damage and gamma ionization damage have been investigated. The typical characterization of the CISs induced by the neutron displacement damage and gamma ionization damage are presented separately. The dark signal degradation in the dark image induced by neutron radiation shows a local increase while the dark signal degradation in the dark image induced by ^60^Co γ-ray radiation shows a uniform increase.

The experimental results show that the degradations of the CISs are very different when the CISs are irradiated by reactor neutron beams and ^60^Co γ-rays in different sequential order. The mean dark signal increase in the CISs induced by reactor neutron radiation at 1 × 10^11^ n/cm^2^ as shown in Figure 10 is 531.65 DN (from 162.34 DN to 693.99 DN). The mean dark signal increase in the CISs induced by reactor neutron radiationat 1 × 10^11^ n/cm^2^ (the CISs were firstly exposed to ^60^Co γ-rays and annealing for 168 h before the reactor neutron radiation.) as shown in Figure 8 is 538.93 DN (from 2112.53 DN to 2651.46 DN). The mean dark signal increase in the CISs induced by reactor neutron radiation has not been influenced by previous ^60^Co γ-ray radiation. However, the mean dark signal increase of the CISs induced by ^60^Co γ-ray radiation has been remarkably influenced by previous reactor neutron radiation. The mean dark signal increase induced by ^60^Co γ-ray radiation at 200 krad(Si) as shown in Figure 8 is 2349.87 DN (from 165.62 DN to 2515.49 DN). The mean dark signal increase only induced by ^60^Co γ-ray radiation at 200 krad(Si)as shown in Figure 10 is 602.07 DN (from 665.40 DN to 1267.47 DN). The mechanisms of the above phenomenon have been explained by the variety in the electric field distribution induced by neutron radiation by TCAD simulation of the pixels in a CIS.

The experimental results have verified the synergistic effects on the 0.18 µm PPD CISs induced by neutron displacement damage and gamma ionization damage. More detailed radiation experiments will be carried out in the future to further research the synergistic effects on the CISs induced by displacement damage and ionization damage.

## Figures and Tables

**Figure 1 sensors-24-01441-f001:**
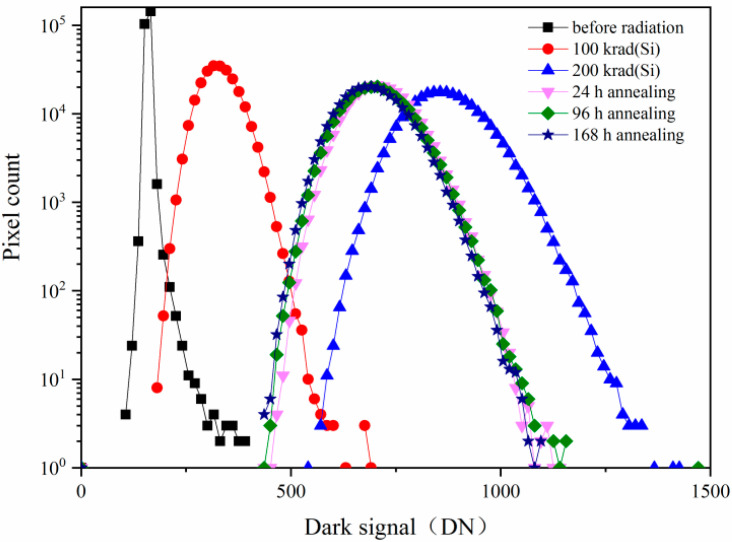
The dark signal distributions induced by ^60^Co γ-ray radiation and annealing at 24 h, 96 h, and 168 h with the same integration time of 16.79 ms.

**Figure 2 sensors-24-01441-f002:**
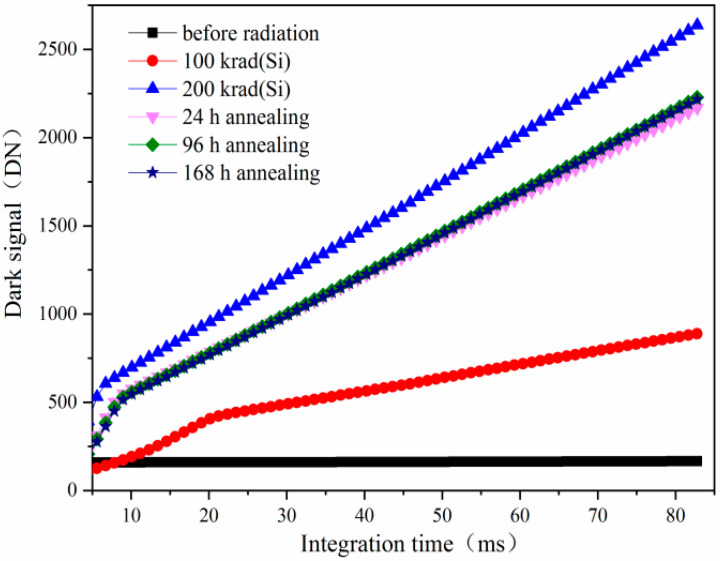
Themean dark signal induced by ^60^Co γ-ray radiation and annealing at 24 h, 96 h, and 168 h versus integration time.

**Figure 3 sensors-24-01441-f003:**
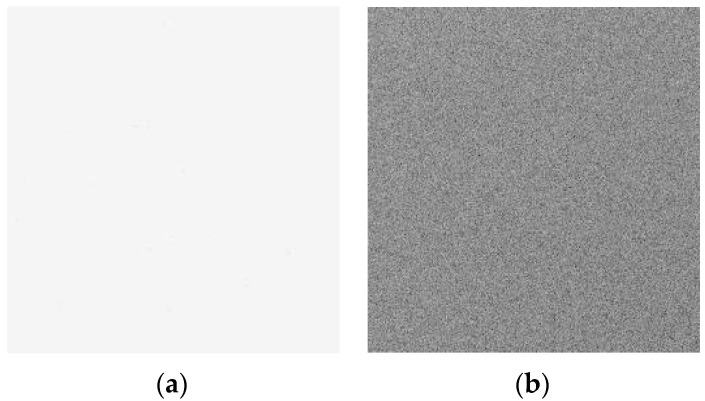
The typical dark images induced by ionization damage (inversion of black and white color): (**a**) before radiation; (**b**) after ^60^Co γ-ray radiation at 200 krad(Si).

**Figure 4 sensors-24-01441-f004:**
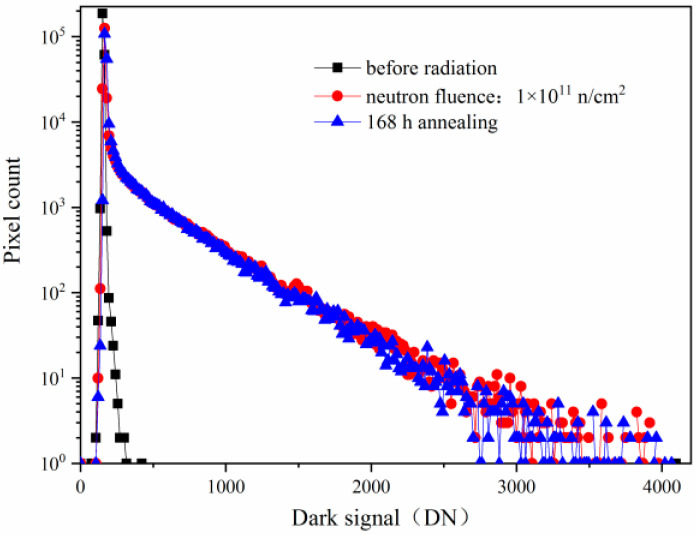
The dark signal distributions induced by reactor neutron radiation and annealing for 168 h with the same integration time of 16.79 ms.

**Figure 5 sensors-24-01441-f005:**
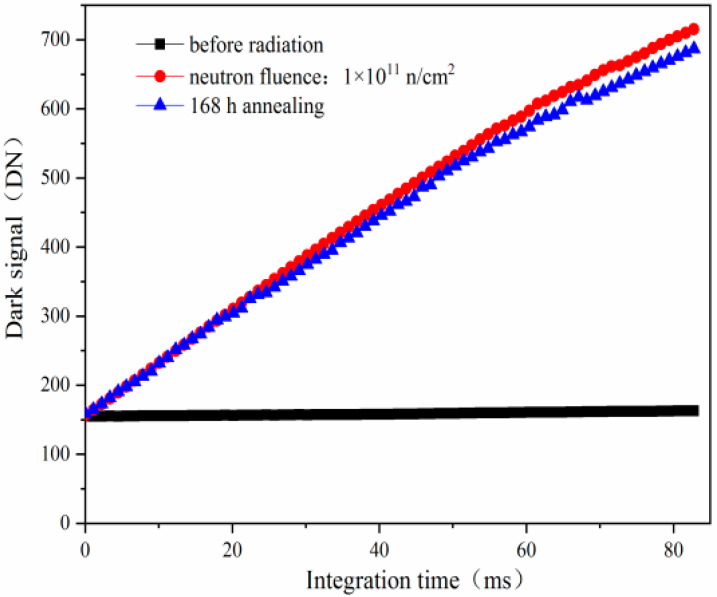
The mean dark signal induced by reactor neutron radiation and annealing for 168 h versus integration time.

**Figure 6 sensors-24-01441-f006:**
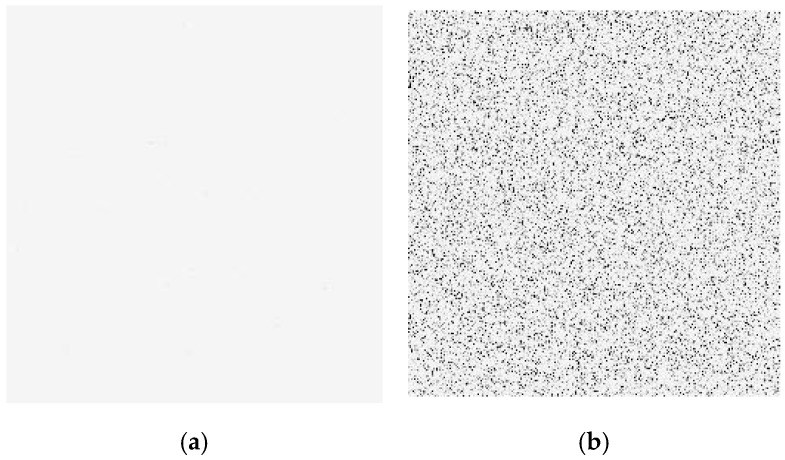
The typical dark images induced by displacement damage (inversion of black and white color): (**a**) before radiation; (**b**) 1 × 10^11^ n/cm^2^ after reactor neutron radiation.

**Figure 7 sensors-24-01441-f007:**
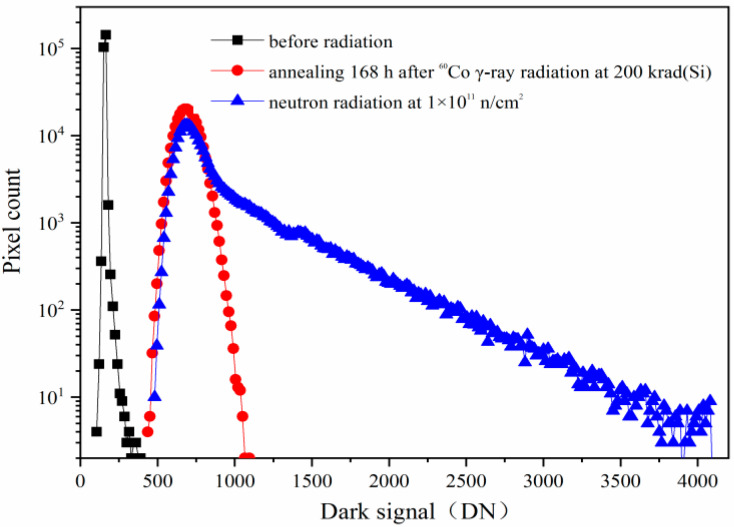
The dark signal distributions before and after radiation: before radiation (black line); ^60^Co γ-ray radiation at 200 krad(Si) after annealing for 168 h (red line); firstly exposed to ^60^Co γ-rays (including annealing for 168 h) and then secondly exposed to reactor neutron beams (blue line).

**Figure 8 sensors-24-01441-f008:**
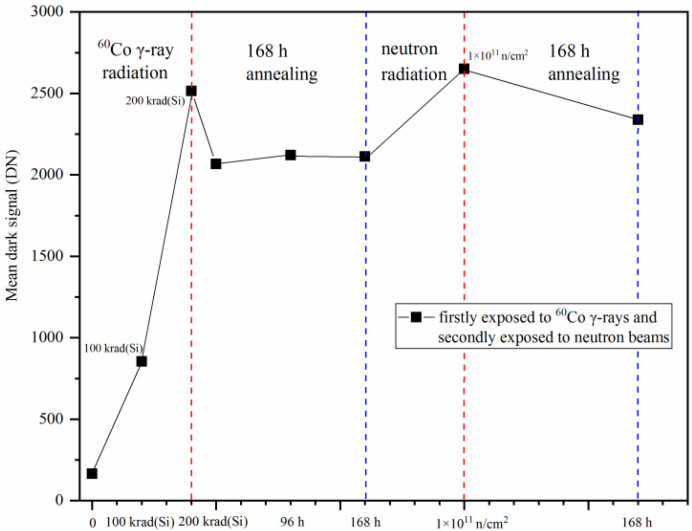
The mean dark signal degradations of the CISs that were firstly exposed to ^60^Co γ-rays and annealing for 168 h, and then were secondly exposed to reactor neutron beams and annealing 168 h.

**Figure 9 sensors-24-01441-f009:**
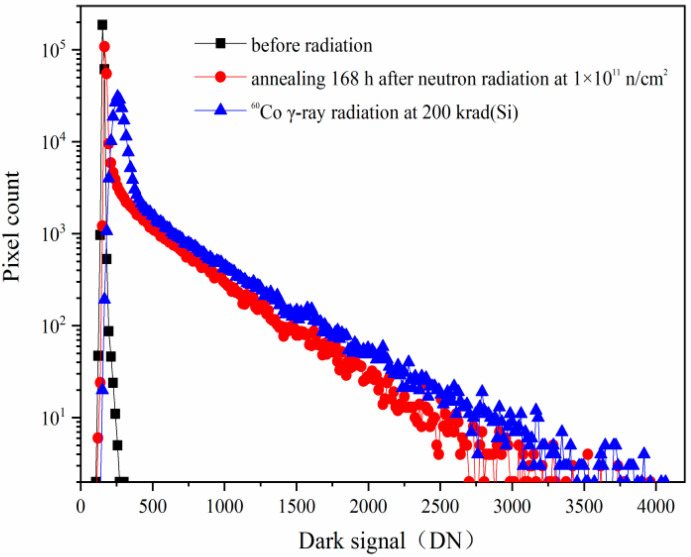
The dark signal distributions before and after radiation: before radiation (black line); reactor neutron beam at 1 × 10^11^ n/cm^2^ after annealing 168 h (red line); firstly exposed to reactor neutron beam (including annealing 168 h) and then were secondly exposed to ^60^Co γ-rays (blue line).

**Figure 10 sensors-24-01441-f010:**
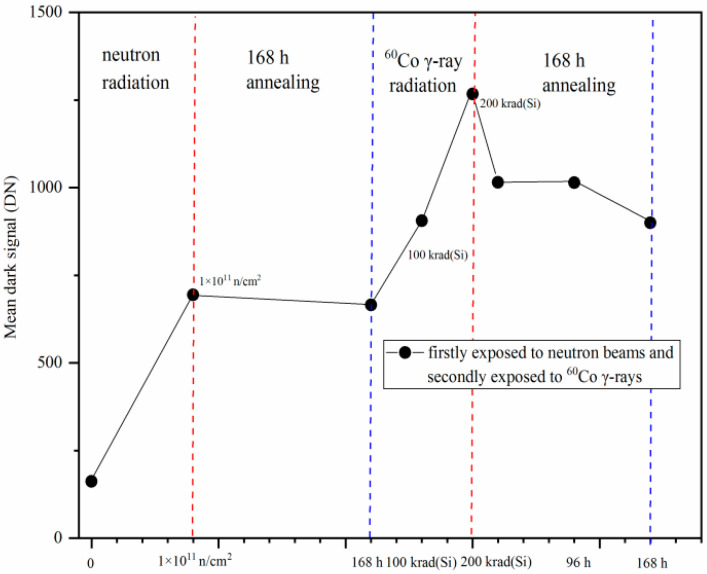
Themean dark signal degradations of the CISs that were firstly exposed to reactor neutron beams and annealing for 168 h, and then were secondly exposed to ^60^Co γ-rays and annealing for 168 h.

**Figure 11 sensors-24-01441-f011:**
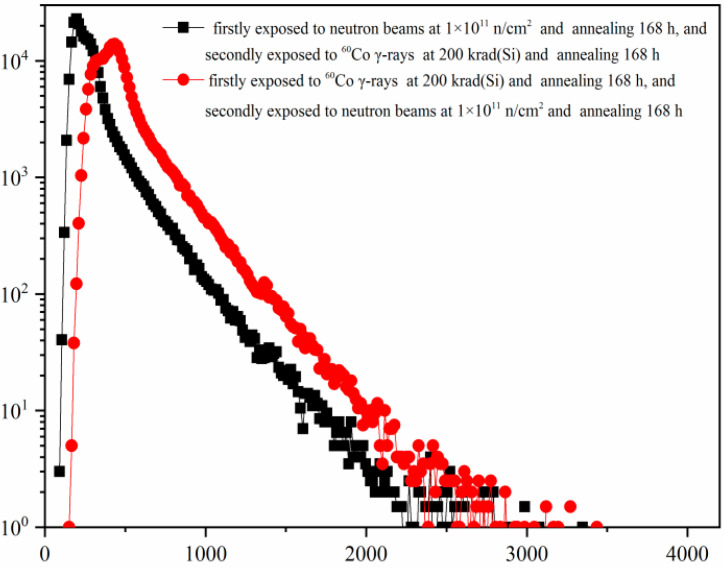
The comparison of the dark signal distributions of the CISs that were irradiated by reactor neutron beams and^60^Co γ-rays with different sequential order.

**Figure 12 sensors-24-01441-f012:**
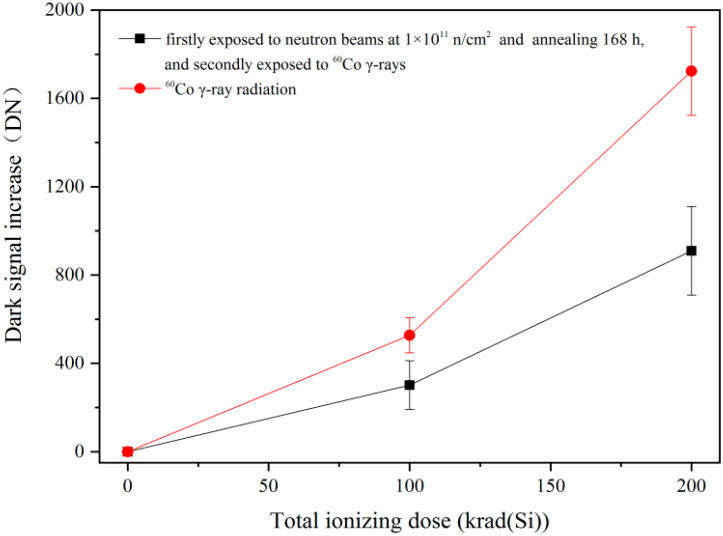
The comparison of the dark signal increase of the CISs that were exposed to ^60^Co γ-rays (one case is that the CISs were firstly irradiated by reactor neutron beams at 1 × 10^11^ n/cm^2^ and annealing 168 for h, and then secondly irradiated by ^60^Co γ-rays).

**Figure 13 sensors-24-01441-f013:**
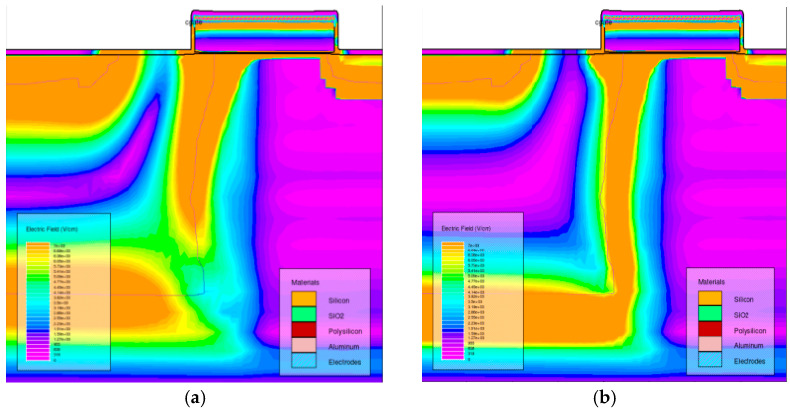
The electric field distribution of the region near the TG and SCR in a CIS:(**a**) before radiation; (**b**) after neutron radiation.

**Figure 14 sensors-24-01441-f014:**
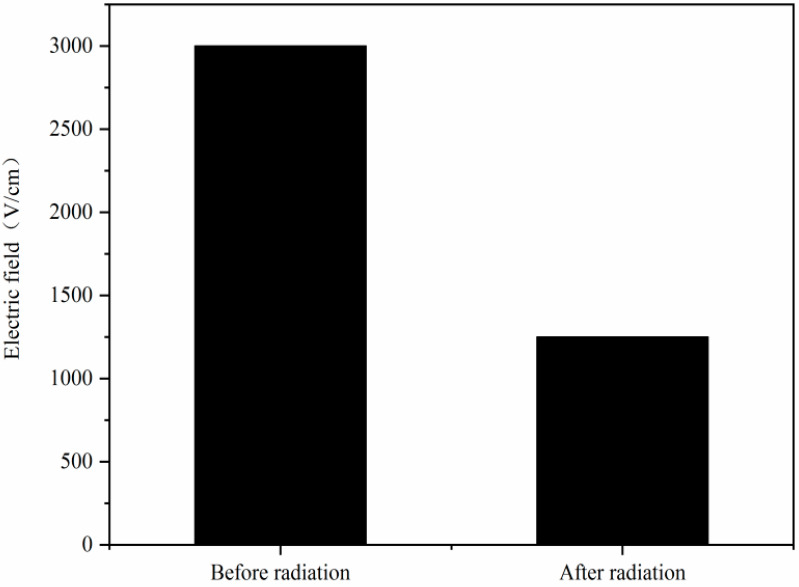
The electric field of the region near the TG and SCR in a CIS before and after neutron radiation.

**Table 1 sensors-24-01441-t001:** Radiation test conditions and samples.

CIS Number	1st Radiation 168 h Annealing		2nd Radiation 168 h Annealing
	Dose Rate (rad(Si)/s)	Total Dose (krad(Si))	Neutron Fluence (n/cm^2^)
A01	50.0	100, 200	1 × 10^11^
A02	50.0	100, 200	1 × 10^11^

**Table 2 sensors-24-01441-t002:** Radiation test conditions and samples.

CIS Number	1st Radiation 168 h Annealing	2nd Radiation 168 h Annealing	
	Neutron Fluence (n/cm^2^)	Dose Rate (rad(Si)/s)	Total Dose (krad(Si))
B01	1 × 10^11^	50.0	100, 200
B02	1 × 10^11^	50.0	100, 200

## Data Availability

Data is unavailable due to privacy.

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
