# Peer review of "Synergistic Radiation Effects in PPD CMOS Image Sensors Induced by Neutron Displacement Damage and Gamma Ionization Damage"

_sensors, 2024, doi:10.3390/s24051441_

Round 1
Reviewer 1 Report
Comments and Suggestions for Authors
This paper studys the synergistic radiation effects in PPD CMOS image sensors. It clearly demonstrates the sequence of applying radiation of gamma ray and neutron can affect the distribution of dark signals in the sensor. The mechanism is further studied by TCAD simulation. The proofs are solid and reasonable. It is good from the academic view point.
However, in the real world, the radiation can happen concurrently. I think it would be great if the authors can explore what happen if the DUT issubject to both gamma ray and neutron radiations at the same time. It would make this paper much more contributive.
Reviewer 2 Report
Comments and Suggestions for Authors
In this manuscript, the authors have investigated the synergistic effects on the 0.18-µm PPD CISs induced by neutron displacement dam-13 age and gamma ionization damage. The corresponding results showed that the mean dark signal increase of the CISs in-18 duced by reactor neutron beams radiation was influenced by previous 60Co γ-ray radia-19 tion. The work may be of some significance, however, in my opinion, there are several issues which need major revisions before publication. Some comments are as following.
1. There is statement "…because the neutron radiation mainly induces the displacement damage while the gamma radiation mainly induces the ionization damage…" in the manuscript. Why? Please provide the corresponding explanation.
2. The characterization of Two samples (A01 and A02) should be provided in the manuscript.
3. The setup of the tested CISs should be added in the manuscript.
4. Fig. 1 shows the dark signal distributions induced by 60Co γ-ray radiation and an-nealing at 24 h, 96 h, 168 h with the same integration time of 16.79 ms. How did the authors determine the time?
5. How is the experimental work in the manuscript compared with other results in literature? The authors should also briefly discuss this.
6. The English writing is poor and there are some obvious grammar errors, which should be polished.
Comments on the Quality of English Language
The English writing is poor and there are some obvious grammar errors, which should be polished.
Round 2
Reviewer 1 Report
Comments and Suggestions for Authors
I have no further questions.
Reviewer 2 Report
Comments and Suggestions for Authors
Since all the comments have been considered and discussed carefully, I would recommend this manuscript for publication.